# Enhanced Latent-Space Adversarial Training for Super-Resolution

Liangbin Xie [1 2]   Zheyuan Li [1 2]   Fanghua Yu [3]   Xinqi Lin [2]   Junhao Zhuang [4]   Jinfan Hu [2]   Jinjin Gu [5]
Jiantao Zhou [1]   Chao Dong [2 6]

## Abstract

Real-world super-resolution (SR) at large upscaling factors (i.e., $\geq 4\times$) remains difficult due to complex real-image degradations. HYPIR, a leading diffusion-based restoration model, performs strongly on many inputs, yet for a non-trivial portion of more challenging cases a single forward step does not fully recover fine-grained details. A naive two-stage cascade improves visual quality, but introduces over-saturation, weak texture details, and high inference latency. To address these issues, this paper proposes HYPIR++. It removes the degradation removal encoder and noise augmentation modules to better preserve fidelity cues from low-quality inputs. Equipped with an optimized latent ConvNeXt and a latent patch discriminator, HYPIR++ supports latent-space adversarial learning for clearer details and more stable local structures. It further shortens text sequences and replaces full attention with sparse neighbor attention, enabling efficient high-resolution inference without tiling. Experiments show that HYPIR++ improves perceptual quality and runs $1.71\times$ faster than HYPIR on large-factor real-world SR.

## 1. Introduction

Image super-resolution (ISR) (Dong et al., 2014; 2016; Zhang et al., 2018b; Kim et al., 2016; Ledig et al., 2017; Wang et al., 2018; Liang et al., 2021; Chen et al., 2023) seeks to recover a high-quality (HQ) image from a single low-quality (LQ) input, which often suffers from a mixture of degradations (Gu et al., 2019; Zhang et al., 2021; Wang et al., 2021) (e.g., noise, blur, JPEG compression). Early approaches relying on GAN training (Ledig et al., 2017; Wang et al., 2018; Zhang et al., 2021; Wang et al., 2021) often introduce unrealistic artifacts (Liang et al., 2022; Xie et al., 2023). The advent of diffusion models (Rombach et al., 2022; Podell et al., 2023; Esser et al., 2024) marked a significant shift. The models (Wang et al., 2024a; Wu et al., 2024c; Lin et al., 2024; Yu et al., 2024) that leverage the diffusion models as prior demonstrated a powerful capability to generate photorealistic details. Yet, their iterative nature, often requiring 20 to 50 sampling steps in inference stage, severely hinders their practical applicability. Consequently, a subsequent line of research (Wang et al., 2024b; Wu et al., 2024b) has focused on distilling these multi-step models into a single-step inference network. Recent work, HYPIR (Lin et al., 2025c), especially its FLUX version, achieves notable restoration performance in a single inference step, by initializing from a pretrained diffusion model and fine-tuning adversarially in pixel space.

In this paper, we focus on real-world SR at large upscaling factors—concretely, factors of at least $4\times$, with $8\times$ being a natural extension. Such large-factor settings are markedly more challenging than small-factor ones (e.g., $2\times$): the input is severely degraded, and a single network must jointly remove these degradations and synthesize substantial high-frequency content. In this regime, HYPIR is already a strong baseline overall, and for many inputs a single forward step already suffices; however, for a non-trivial portion of more challenging cases, a single step does not fully restore fine-grained details, as exemplified by Fig. 1(b). We note that this is not a deficiency unique to HYPIR, but rather reflects the intrinsic difficulty of recovering severe, composite degradations together with detail-enhancing upsampling in one shot. Motivated by the observation that allocating more computation at inference often improves model performance (Muennighoff et al., 2025; Ma et al., 2025), a natural remedy for these challenging cases is to decompose the task into simpler sequential sub-tasks. A naive two-stage cascade of HYPIR (Fig. 1(c)) confirms this intuition and improves perceptual quality on such inputs. Yet, this naive cascade brings three practical drawbacks: 1) over-saturation accumulated across stages, 2) limited fine-

---

[1]State Key Laboratory of Internet of Things for Smart City, University of Macau [2]Shenzhen Institutes of Advanced Technology, Chinese Academy of Sciences [3]The University of Hong Kong [4]Tsinghua University [5]INSAIT, Sofia University "St. Kliment Ohridski" [6]Shenzhen University of Advanced Technology. Correspondence to: Jiantao Zhou <jtzhou@um.edu.mo>, Chao Dong <chao.dong@siat.ac.cn>.

*Proceedings of the 43rd International Conference on Machine Learning*, Seoul, South Korea. PMLR 306, 2026. Copyright 2026 by the author(s).

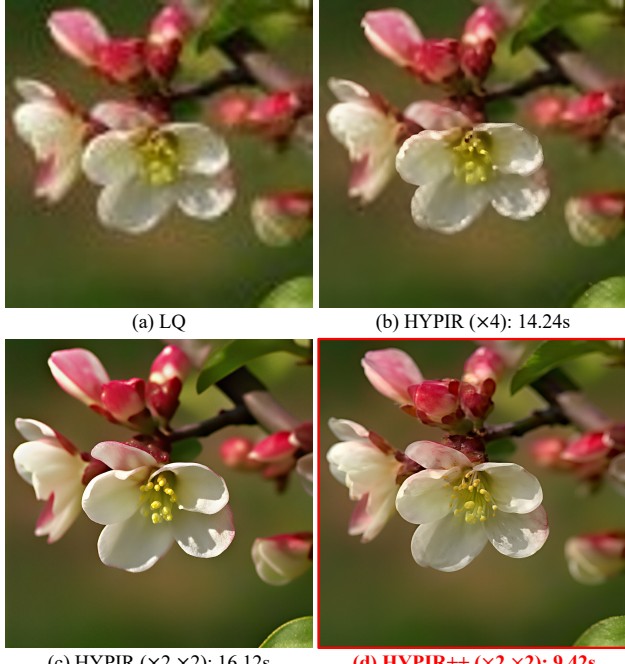

(a) LQ      (b) HYPIR (×4): 14.24s

(c) HYPIR (×2 ×2): 16.12s      **(d) HYPIR++ (×2 ×2): 9.42s**

*Figure 1.* Visual comparison of ×4 SR results. (a) LQ input; (b) HYPIR run as a single ×4 step; (c) HYPIR run as a ×2∘ ×2 cascade; (d) HYPIR++ run as a ×2∘ × 2 cascade. Per-image inference latency is shown next to each method.

grained details, and 3) prohibitive inference latency caused by full attention and the tiling operation required to fit GPU memory. These drawbacks make a naive cascaded HYPIR less attractive in practice for large-factor real-world SR.

In this work, we introduce HYPIR++, a framework tailored for two-stage cascaded large-factor SR that addresses the above three drawbacks. First, we trace the origin of over-saturation to the FLUX model itself, which tends to generate oversaturated results (Xue et al., 2025; Shen et al., 2025b). To eliminate this generative bias, HYPIR++ removes the degradation removal encoder and noise augmentation modules. This forces the model to adhere more closely to the fidelity cues (e.g., color, saturation) provided by the low-quality (LQ) input, alleviating the over-saturation problem that is otherwise amplified by cascading.

Second, to enhance fine-grained detail generation, we move the adversarial framework into the latent space. Since no efficient pre-trained feature extractor exists for this space, we design and distill a specialized latent ConvNeXt (Liu et al., 2022b) as the discriminator backbone. To strengthen local structure, a complementary latent patch discriminator operates on randomly cropped latent patches and emits dense realness maps for localized supervision.

Finally, to alleviate inference latency, we reduce the input text-token length (e.g., from 512 to 128) and replace full attention with our proposed sparse neighbor attention (SNA). SNA efficiently limits attention computations to spatially

neighboring blocks and selects the top-k most relevant blocks, while preserving global text-image interactions. By leveraging this neighbor-based sparsity pattern, HYPIR++ generalizes well across different resolutions, supports direct processing of large images, and eliminates the need for slow block-based tiling.

In summary, our contributions are as follows:

- We propose HYPIR++, a cascade-aware framework for large-factor ($\geq 4\times$) real-world SR. By removing the degradation removal encoder and noise augmentation, HYPIR++ preserves fidelity cues across cascaded stages and alleviates the over-saturation that is otherwise amplified by cascading.

- We design a latent ConvNeXt and conduct latent-space GAN training for HYPIR++, enhancing fine-grained details. Furthermore, a latent patch discriminator is introduced to help restore local structures.

- We develop several strategies to improve efficiency, including reducing the text sequence length and replacing full attention with the proposed sparse neighbor attention, which enables direct processing of high-resolution images without block-based tiling.

- Under the same two-stage cascaded setting, HYPIR++ delivers higher perceptual quality than cascaded HYPIR with a $1.71\times$ speedup, providing a practical solution for large-factor real-world SR.

## 2. Related Work

### 2.1. Image Super-Resolution (ISR)

Image super-resolution aims to recover a high-quality image from its degraded counterparts. Classical approaches (Dong et al., 2014; 2016; Zhang et al., 2018b; Kim et al., 2016; Ledig et al., 2017; Wang et al., 2018; Liang et al., 2021; Chen et al., 2023) focus on a specific bicubic degradation. Concurrently, the problem scope has expanded from addressing single, well-defined degradations (e.g., bicubic down-sample, noise, blur, JPEG compression) to tackling complex, composite artifacts present in real-world scenarios (Gu et al., 2019; Zhang et al., 2021; Wang et al., 2021). Beyond pure fidelity, recent works focus on enhancing perceptual quality by leveraging GAN (Wang et al., 2021; Yang et al., 2021) and diffusion generative priors (Wang et al., 2024a; Wu et al., 2024c; Lin et al., 2024; Yu et al., 2024; Yue et al., 2025b; Dong et al., 2025; Wu et al., 2024b; Lin et al., 2025c).

### 2.2. Efficient Diffusion-based ISR

While diffusion models have demonstrated superior performance, their substantial computational requirements pose

a significant barrier to practical usage. To reduce inference latency, two primary research directions have emerged. The first is to reduce the number of inference steps. This is typically achieved through techniques such as rectified flows (Liu et al., 2022a; Yin et al., 2024b; Liu et al., 2023), score distillation (Lin et al., 2025b; Zhang et al., 2024b), and adversarial training paradigms (Yin et al., 2024a; Wang et al., 2023b). In the ISR domain, many methods adopt these strategies and yield models capable of single-step inference, including OSEDiff (Wu et al., 2024b), SinSR (Wang et al., 2024b), TSD-SR (Dong et al., 2025), FlashVSR (Zhuang et al., 2025), and HYPIR (Lin et al., 2025c). The second direction is to mitigate the quadratic complexity of the self-attention mechanism, often by leveraging sparsity or local attention operations. This approach has been predominantly explored in text-to-image (Shen et al., 2025a) and text-to-video generation (Xi et al., 2025; Yang et al., 2025), with limited application in ISR. In this work, built upon HYPIR, we integrate the sparse neighbor attention mechanism to maintain high computational efficiency. In addition, since a single forward step does not always suffice for the most challenging large-factor inputs, we adopt a two-stage cascade with optimization performed in the latent space to further improve restoration quality in large-factor real-world SR.

### 2.3. High-Resolution Generation

Ultra-high-resolution generation remains challenging; representative methods adopt an overlapping tiling strategy (Bar-Tal et al., 2023; Lee et al., 2023) but incur extra computational cost on overlapping regions. Current ISR methods (Yu et al., 2024; Lin et al., 2025c) likewise resort to tiling for large-scale images, which typically involves dividing the image into patches, processing them individually, and stitching the results, again at a significant computational overhead. In our work, the sparse neighbor attention mechanism enables our model to generalize across different resolutions and process large images directly, eliminating the need for such tiling operations.

### 3. Preliminaries: HYPIR

In Fig. 2 (a), we detail the training pipeline of HYPIR. First, the low-quality (LQ) image $X_L$ is processed by a fine-tuned degradation removal encoder to yield $x_L$. Subsequently, $x_L$ is corrupted with random noise and patchified into image tokens $z'_L$. Concurrently, a detailed text prompt is encoded by the T5 text encoder (Raffel et al., 2020) into 512-dimensional text tokens, $z_c$. Both $z'_L$ and $z_c$ are then fed into the DiT (FLUX) model, which is equipped with LoRA parameters. Finally, the predicted latent, $\hat{x}_H$, is passed through the VAE decoder to generate the final restored image, $\hat{X}_H$.

To adapt the LoRA-equipped DiT (FLUX) for the image restoration task, HYPIR employs the GAN training strat-

egy (Ledig et al., 2017; Wang et al., 2018). The framework computes a distance loss ($L_2$), a perceptual loss ($L_{\text{LPIPS}}$), and a GAN loss ($L_{\text{GAN}}$) between the restored output $\hat{X}_H$ and the high-quality (HQ) ground truth $X_H$. The GAN loss ($L_{\text{GAN}}$) is computed based on features extracted from a pretrained ConvNeXt (Liu et al., 2022b) and mapped to a scalar prediction by a lightweight head module. While the degradation removal encoder is pre-trained in a separate stage, only the LoRA parameters and the head of the discriminator are updated during GAN fine-tuning.

### 4. Methodology: HYPIR++

HYPIR++ targets the two-stage cascaded large-factor SR scenario ($\geq 4\times$ in this paper). *At inference*, the total upscaling factor is decomposed into two sequential sub-factors, and a single generator is invoked twice with shared weights (e.g., a $\times 4$ task is realized as two consecutive $\times 2$ passes; Fig. 3). A single forward step does not always recover sufficient detail for the most challenging inputs, while a naive cascade—though effective in restoration quality—suffers from over-saturation, weakened fine-grained details, and prohibitive inference latency—issues that HYPIR++ resolves with a cascade-aware design that revisits both the training pipeline and the inference procedure.

The training framework of HYPIR++ is illustrated in Fig. 2 (b), and the corresponding cascaded inference pipeline is shown in Fig. 3. Under this two-stage inference setting, we make three training-side changes that together yield a more cascade-friendly model. First, we eliminate over-saturation by removing the degradation removal encoder and noise augmentation modules (Sec. 4.1). Second, we enhance fine-grained detail generation through latent-space adversarial training with our designed latent ConvNeXt and latent patch discriminator (Sec. 4.2). Third, by shortening the text sequence length and incorporating sparse neighbor attention (Sec. 4.3), HYPIR++ supports tiling-free inference, which is especially valuable in a two-stage cascade where any per-stage cost is doubled.

### 4.1. Enhancing Fidelity to LQ Input

As illustrated in Fig. 1, the results from HYPIR exhibit over-saturation, which becomes more pronounced when the model is cascaded multiple times. This over-saturation originates from the FLUX model's inherent tendency to produce overly saturated outputs (Xue et al., 2025; Shen et al., 2025b). In HYPIR, the degradation removal encoder and noise augmentation reinforce this bias by corrupting the fidelity cues in the LQ input, leading the model to rely more on FLUX's generative prior. To address this, HYPIR++ removes both modules, allowing the model to better preserve the input's fidelity cues.

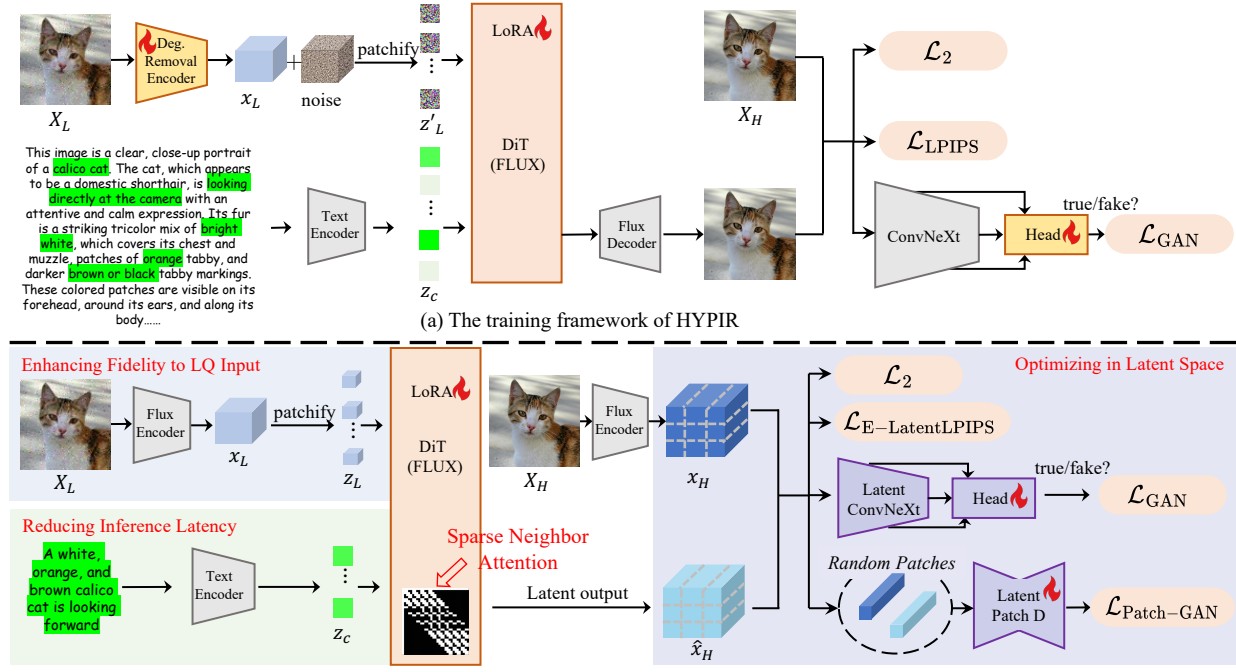

*Figure 2.* Training framework comparison. HYPIR (a) employs a degradation removal encoder with noise augmentation; HYPIR++ (b) removes these modules and performs adversarial training in the FLUX latent space with sparse neighbor attention, enabling tiling-free high-resolution inference.

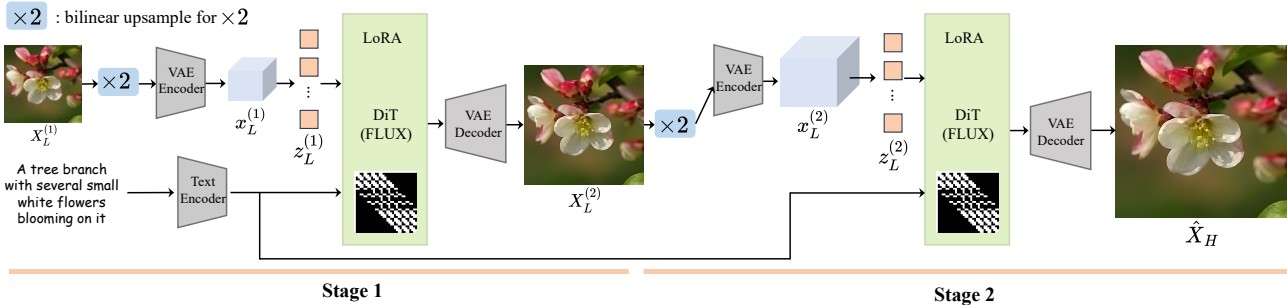

*Figure 3.* Two-stage cascaded inference pipeline of HYPIR++, shown for $\times 4$ SR with sub-factors $s_1 = s_2 = 2$. Both stages share the same generator: the stage-$i$ input image $X_L^{(i)}$ is bilinearly upsampled by $\times 2$, encoded into the latent $x_L^{(i)}$, patchified into tokens $z_L^{(i)}$, and processed by the LoRA-equipped DiT (FLUX) with text conditioning. The decoded output of stage 1, $X_L^{(2)}$, feeds into stage 2, whose decoded output is the final restoration $\hat{X}_H$. The total upscaling factor equals $s_1 \cdot s_2$.

## 4.2. Optimizing in Latent Space

In HYPIR, all losses are computed in the image space. This design requires the gradients to backpropagate through the frozen VAE decoder to update the DiT's LoRA weights, thereby providing only indirect supervision to the DiT. Therefore, HYPIR++ transitions all loss calculations directly to the latent space (i.e., before the VAE decoder), enabling direct gradient flow to the DiT.

**Latent ConvNeXt.** While recent works (Lin et al., 2025a; Kang et al., 2024) suggest using diffusion models as latent-space discriminators, we explore an alternative network due to VRAM considerations. The FLUX model itself already consumes significant VRAM, making the addition of another large diffusion-based discriminator computationally prohibitive. Following HYPIR, we opt for a pre-trained vision model (e.g., ConvNeXt (Liu et al., 2022b)) as the backbone of the discriminator. Since such a model is pre-trained on RGB images and cannot be directly applied to latent-space features, we train a specialized latent ConvNeXt to serve as the feature extractor. To accelerate convergence, we employ knowledge distillation, designating a pre-trained ConvNeXt as the teacher model (Fig. 4 (a)) and the latent ConvNeXt as the student model. The training objective is the squared $L_2$ loss (MSE):

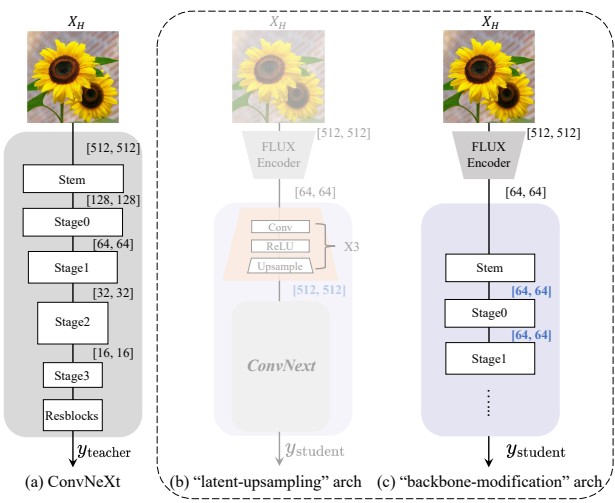

*Figure 4.* Architecture of the latent ConvNeXt. (a) Teacher–student distillation setup; (b)(c) two candidate backbones (latent-upsampling vs. backbone-modification).

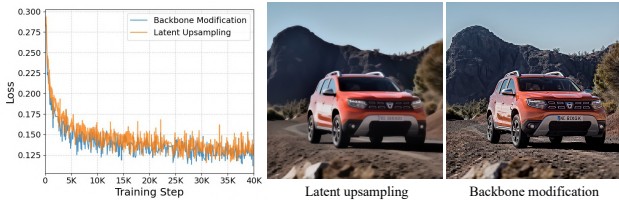

*Figure 5.* Comparison between the two latent ConvNeXt variants in distillation loss and downstream restoration quality.

$$L_{\text{distill}} = \|y_{\text{teacher}} - y_{\text{student}}\|_2^2 \tag{1}$$

Given that the FLUX encoder maps a 512×512 image to a 64×64 latent, we design two straightforward architectures for the latent ConvNeXt. The first architecture, shown in Fig. 4 (b), is a "latent-upsampling" variant: it reuses the original ConvNeXt backbone, but employs a pre-processing module (three Conv-ReLU-Upsample blocks) to upscale the $64 \times 64$ latent to $512 \times 512$. The second architecture, shown in Fig. 4 (c), is a "backbone-modification" variant: it directly modifies the ConvNeXt backbone by removing the downsampling operations within its initial Stem and Stage1 modules, enabling it to process the 64×64 latent. During the distillation phase, we observe that both architectures converge successfully (shown in the curve of Fig. 5). However, as illustrated by the two restoration results in Fig. 5, when integrated into the GAN training loop, only the "backbone-modification" version can effectively guide the generator to produce sharper and richer details.

**Latent patch U-Net discriminator.** While the discriminator equipped with the distilled latent ConvNeXt is effective at assessing global consistency, its holistic focus can overlook fine-grained local structures. To address this specific limitation, we introduce a complementary, patch-based dis-

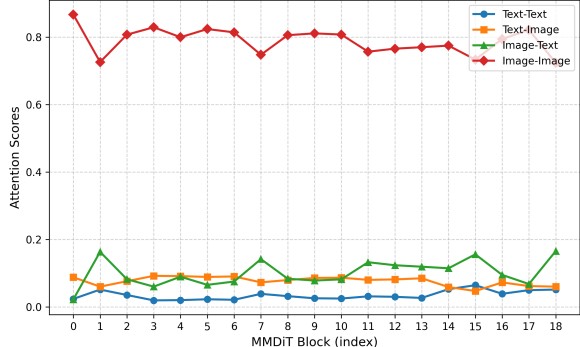

*Figure 6.* Distribution of attention scores within the FLUX MMDiT blocks: image-to-image interaction dominates (∼80%), motivating sparse neighbor attention on image tokens.

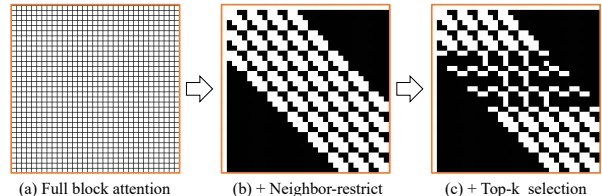

*Figure 7.* The generation process of the sparse neighbor attention map, where 1 marks blocks included in attention.

criminator. Specifically, given the 64×64 latent map, we randomly crop multiple small latent patches (i.e., 8×8). These patches are then fed into a lightweight U-Net discriminator with skip-connections, which outputs a dense realness map for each patch. These maps provide per-pixel feedback to the generator and thereby enforce local realism.

**Training objectives.** Following HYPIR, HYPIR++ also employs a GAN training strategy. For the perceptual similarity calculation, we directly adopt the E-LatentLPIPS (Kang et al., 2024) loss, given its demonstrated effectiveness in measuring perceptual quality between latents in the FLUX space. The simplified objective function can be written as:

$$L_{\text{total}} = \lambda_1 L_2 + \lambda_2 L_{\text{E−LatentLPIPS}} + \lambda_3 L_{\text{GAN}} + \lambda_4 L_{\text{patch−GAN}}, \tag{2}$$

where the weights are empirically set to $\lambda_1 = 1$, $\lambda_2 = 5$, $\lambda_3 = 1$, $\lambda_4 = 1$.

### 4.3. Reducing Inference Latency

**Reducing text sequence length in FLUX.** HYPIR++ shortens the input text prompt, reducing the text-token sequence length (e.g., $512 \to 128$) to lower computational overhead. This design is motivated by our analysis of attention scores within the FLUX MMDiT blocks (Labs, 2024): as plotted in Fig. 6, image-to-image interaction accounts for ∼80% of the total attention weight, so trimming text-token granularity has negligible impact on restoration quality.

**Sparse neighbor attention (SNA).** The full attention mechanism in DiT presents a significant computational bottleneck. Following (Guo et al., 2024), SNA first partitions the tokens into non-overlapping spatial blocks, each containing 128 tokens. For the text modality, which we have already reduced to 128 tokens, we treat the entire sequence as a single block. For the image modality, assuming an original sequence length of $L$, the queries (Q) and keys (K) are partitioned into non-overlapping spatial blocks of size (8, 16). These are then reshaped into $(B, N, 128, C)$, where the total block number $N = L/128$. When the input resolution is 1024×1024, there are 33 blocks (shown in Fig. 7 (a)). Given that the text modality constitutes a single block, SNA applies its neighbor-based sparsity only to the image blocks. As illustrated in Fig. 7 (b, c), the blocks in the first row and first column—corresponding to the text modality—are always retained in the subsequent attention.

The sparsity mechanism is applied in two phases. First, each image query is restricted to attending only to keys and values within a neighboring region (shown in Fig. 7 (b)), ensuring a consistent receptive field across different resolutions. Second, for each image block, average pooling is applied to its Q and K tensors to obtain compact, block-level features, which are then used to compute a coarse block-to-block attention map. As shown in Fig. 7 (c), the top-$k$ most relevant image-to-image block pairs are selected. Finally, full $128 \times 128$ attention is applied—using the original (Q, K, V) tensors—only to these selected neighboring blocks.

# 5. Experiments

## 5.1. Experimental Settings

**Datasets and metrics.** The overall training dataset comprises approximately 20 million high-quality image patches with associated textual descriptions and an additional 70 thousand face images. To synthesize degraded inputs, we adopt the degradation pipeline from Real-ESRGAN (Wang et al., 2021; Lin et al., 2025c). Our evaluation is conducted on three real-world datasets (DRealSR (Wei et al., 2020), RealSR (Cai et al., 2019) and RealPhoto60 (Yu et al., 2024)).

**Evaluation metrics.** We evaluate the synthetic datasets using the full-reference metrics: PSNR, SSIM (Wang et al., 2004), and LPIPS (Zhang et al., 2018a). To more comprehensively evaluate perceptual quality, we employ several no-reference image quality assessment (IQA) metrics: NIQE (Mittal et al., 2012), MANIQA (Yang et al., 2022), MUSIQ (Ke et al., 2021), CLIP-IQA (Wang et al., 2023a), and DeQA (You et al., 2025).

**Implementation details.** We set the LoRA rank to 64 and the total batch size to 8, and train the generator, head, and latent patch U-Net with Adam (Kingma, 2014) at a learning rate of $1 \times 10^{-5}$ for 50K iterations on 8 NVIDIA A6000

GPUs. Training images are first cropped to 1024×1024 and then resized to 512×512 to fit VRAM. At inference, we process large images directly without tiling. For SNA, the neighbor range is 3, and the top-$k$ ratio is set to ∼50% for inputs up to 1024×1024 and ∼33% otherwise.

## 5.2. Comparison with Existing Methods

We compare HYPIR++ with the latest state-of-the-art methods, including Real-ESRGAN (Wang et al., 2021), StableSR (Wang et al., 2024a), DiffBIR (Lin et al., 2024), InvSR (Yue et al., 2025a), SeeSR (Wu et al., 2024c), TS-DSR (Dong et al., 2025), S3Diff (Zhang et al., 2024a), OSEDiff (Wu et al., 2024a), and HYPIR (FLUX) (Lin et al., 2025c). Consistent with the scope of this paper, our evaluation is conducted on the large-factor regime ($\geq 4\times$) where a two-stage cascade is genuinely beneficial: ×4 SR on RealPhoto60 and ×8 SR on RealSR and DRealSR.

For a fair comparison, each method runs in its recommended inference mode rather than being forced into a uniform template. Concretely, multi-step diffusion methods (e.g., StableSR, DiffBIR, SUPIR, SeeSR, InvSR) and GAN-based methods (e.g., Real-ESRGAN) follow their default settings, since the former already scale inference compute through their sampling steps and the latter is deterministic. Single-step diffusion-based methods (OSEDiff, TSDSR, S3Diff, HYPIR, and HYPIR++) lack a native multi-step option, and a single forward step is often insufficient at large factors; two-stage cascading is the most natural way to allocate additional inference compute to these methods, and we evaluate this entire family under the two-stage cascaded setting.

**Quantitative comparison.** Table 1 reports ×8 SR results on RealSR and DRealSR. HYPIR++ improves over HYPIR on the majority of perceptual metrics and consistently ranks within the top two among all competing methods. Table 2 further presents ×4 SR on RealPhoto60 (from $512 \times 512$ to $2048 \times 2048$), where HYPIR++ likewise remains competitive on most perceptual metrics. Since no-reference IQA metrics alone (Yu et al., 2024; Xie et al., 2023) cannot fully evaluate generative restoration, we complement the metric-based evaluation with the user study reported next.

**Qualitative comparison.** We present qualitative comparisons in Fig. 8 and Fig. 9. From Fig. 8, it can be observed that many methods struggle to effectively remove degradations or generate convincing details. In contrast, HYPIR++ removes complex degradations and recovers fine-grained details while preserving the identity of the input. Notably, OSEDiff produces results with visible color shifts from the input. As shown in the first set of results in Fig. 9, compared with other methods, HYPIR++ faithfully recovers the digits "6, 7, 8, 9, and 10." In comparison with SUPIR, HYPIR++ is more effective at removing degradations. Relative to HYPIR, HYPIR++ exhibits more natural and well-balanced

| Datasets | Metrics | RealESRGAN | StableSR | DiffBIR | SUPIR | SeeSR | TSDSR | S3Diff | OSEDiff | HYPIR (FLUX) | HYPIR++ (FLUX) |
|---|---|---|---|---|---|---|---|---|---|---|---|
| | NIQE ↓ | 5.132 | 6.459 | 5.391 | 5.965 | 5.357 | **4.577** | 5.530 | 5.330 | 5.246 | 4.960 |
| | MUSIQ | 69.59 | 73.26 | 72.86 | 70.30 | 73.07 | 73.49 | 71.76 | **73.76** | 71.92 | 73.16 |
| RealSR (×8) | MANIQA | 0.500 | 0.617 | 0.569 | 0.529 | 0.577 | 0.618 | 0.514 | 0.622 | 0.583 | **0.635** |
| | CLIP-IQA | 0.541 | 0.660 | 0.636 | 0.593 | 0.632 | **0.734** | 0.619 | 0.692 | 0.679 | 0.704 |
| | DeQA | 3.475 | 3.713 | 3.885 | 3.385 | 3.862 | 3.869 | 3.817 | **4.036** | 3.837 | 4.028 |
| | NIQE ↓ | 4.852 | 6.056 | 5.149 | 6.055 | 5.206 | **4.384** | 5.109 | 5.332 | 5.139 | 4.848 |
| | MUSIQ | 70.29 | 72.64 | 72.73 | 71.09 | 73.36 | 73.38 | 72.72 | **74.04** | 73.28 | 73.56 |
| DRealSR (×8) | MANIQA | 0.500 | 0.615 | 0.568 | 0.552 | 0.583 | 0.628 | 0.533 | 0.627 | 0.607 | **0.658** |
| | CLIP-IQA | 0.552 | 0.682 | 0.632 | 0.603 | 0.645 | **0.729** | 0.649 | 0.725 | 0.707 | 0.724 |
| | DeQA | 3.556 | 3.726 | 3.870 | 3.395 | 3.887 | 3.950 | 3.922 | 4.073 | 3.972 | **4.150** |

*Table 1.* Quantitative comparison on the large-factor ×8 SR task ($128 \times 128 \rightarrow 1024 \times 1024$) on the RealSR and DRealSR datasets, with the best/second-best results in **bold**/underlined.

| Datasets | Metrics | RealESRGAN | StableSR | DiffBIR | InvSR | SeeSR | TSDSR | S3Diff | OSEDiff | HYPIR (FLUX) | HYPIR++ (FLUX) |
|---|---|---|---|---|---|---|---|---|---|---|---|
| | NIQE ↓ | 5.464 | 4.474 | 4.319 | 6.723 | 3.611 | 3.887 | **3.553** | 4.044 | 3.988 | 3.844 |
| | MUSIQ | 61.41 | 68.20 | 64.85 | 47.59 | 73.79 | 70.17 | 72.99 | **74.73** | 72.89 | 73.44 |
| RealPhoto60 (×4) | MANIQA | 0.468 | 0.499 | 0.441 | 0.291 | 0.568 | 0.579 | 0.569 | 0.624 | 0.610 | **0.627** |
| | CLIP-IQA | 0.529 | 0.657 | 0.590 | 0.392 | 0.732 | 0.670 | 0.762 | 0.796 | 0.796 | **0.806** |
| | DeQA | 3.128 | 3.509 | 3.446 | 2.546 | 4.077 | 3.520 | 4.168 | 4.208 | 4.052 | **4.252** |

*Table 2.* Quantitative comparison on the ×4 SR task ($512 \times 512 \rightarrow 2048 \times 2048$) on the real-world RealPhoto60 dataset, with the best/second-best results in **bold**/underlined.

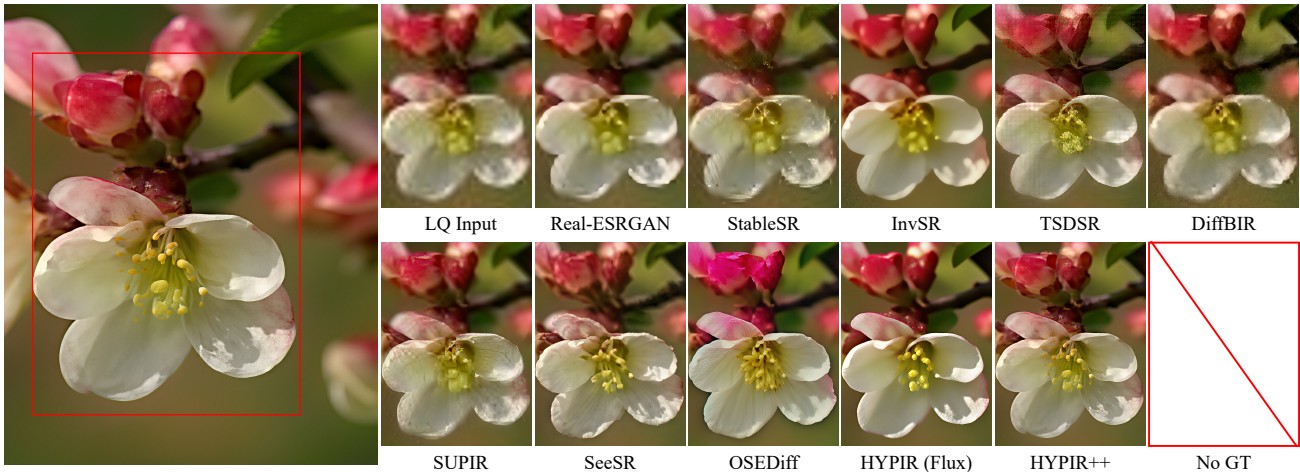

*Figure 8.* Visual comparison on RealPhoto60: HYPIR++ jointly removes complex degradations and restores fine-grained details.

color saturation. For the second set of results, in regions such as petals and leaves, HYPIR++ restores cleaner and sharper details than other methods.

### 5.3. User Study

Since IQA metrics do not fully capture perceptual quality (Yu et al., 2024; Xie et al., 2023), we conduct two user studies: one comparing HYPIR and HYPIR++, and another comparing HYPIR++ with current state-of-the-art methods. Each study uses 20 image groups and is conducted with 45 participants. In the first study, participants are asked to select the result that looks visually better; in the second, participants are asked to select the top-2 most visually pleasing results for each group. As shown in Fig. 10, most participants prefer HYPIR++ over HYPIR, and HYPIR++ is also more frequently preferred than other competing methods.

### 5.4. Ablation Studies

**Effect on over-saturation.** Fig. 11 presents a qualitative comparison between HYPIR++ and HYPIR on a real-world ×4 SR task. By removing the degradation removal encoder and noise augmentation modules, HYPIR++ alleviates the over-saturation issue in the original HYPIR results.

**Effectiveness of latent space optimization.** We present a quantitative and qualitative comparison of the optimization spaces in Tab. 3 and Fig. 12, respectively. Quantitatively, Tab. 3 shows that transitioning the loss calculations to the FLUX latent space clearly outperforms the image space baseline across most metrics. Furthermore, incorporating the latent patch U-Net discriminator further boosts performance on most metrics. Qualitatively, the model optimized in image space produces results that suffer from a lack of fine-grained detail and exhibit global grid-like artifacts. While transitioning to the latent space unleashes the model's detail generation capabilities, it struggles with local struc-

| | | | | | |
|---|---|---|---|---|---|
| LQ | RealESRGAN | StableSR | InvSR | TSDSR | DiffBIR |
| SUPIR | SeeSR | S3Diff | OSEDiff | HYPIR | HYPIR++ |
| LQ | RealESRGAN | StableSR | InvSR | TSDSR | DiffBIR |
| SUPIR | SeeSR | S3Diff | OSEDiff | HYPIR | HYPIR++ |

*Figure 9.* Visual comparisons on the DRealSR (top) and RealSR (bottom) datasets at ×8 SR.

| Optimization Space | NIQE ↓ | MUSIQ | MANIQA | CLIP-IQA | DeQA |
|---|---|---|---|---|---|
| Image Space | 5.957 | 73.267 | 0.639 | 0.792 | 3.998 |
| Latent Space | 4.155 | **74.712** | 0.654 | 0.827 | 4.209 |
| Latent Space + Patch U-Net | **3.977** | 74.382 | **0.657** | **0.831** | **4.247** |

*Table 3.* Ablation on the optimization space (RealPhoto60).

| Resolution | Text Tokens | Time ↓ | Speedup | MUSIQ | MANIQA | CLIP-IQA | DeQA |
|---|---|---|---|---|---|---|---|
| 512 × 512 | **128** | **0.90** | **1.33 ×** | 75.176 | 0.744 | 0.849 | 4.220 |
| | 512 | 1.20 | 1.00 × | 75.157 | 0.745 | 0.851 | 4.228 |
| 1024 × 1024 | **128** | **3.06** | **1.12 ×** | 74.382 | 0.657 | 0.831 | 4.247 |
| | 512 | 3.44 | 1.00 × | 74.499 | 0.659 | 0.832 | 4.232 |

*Table 4.* Ablation study on text token length on the RealPhoto60 dataset, with comparisons across varying output resolutions.

ture (e.g., glasses). The addition of the latent patch U-Net can produce more reasonable structures.

**Inference latency discussion.** Tab. 4 shows that shortening the text tokens from 512 to 128 yields a speedup at both 512×512 and 1024×1024 resolutions with negligible quality loss. Tab. 5 further compares full attention with SNA: SNA matches the full-attention baseline in quality while enabling tiling-free processing of high-resolution inputs, delivering up to a 1.71× speedup at the ×4 setting.

| RealPhoto60 | Types | Time (s) ↓ | Speedup | NIQE ↓ | MUSIQ | MANIQA | CLIP-IQA | DeQA |
|---|---|---|---|---|---|---|---|---|
| × 4 | Full attention | 16.12 | 1.00 × | 3.846 | 73.913 | 0.626 | 0.805 | 4.201 |
| | SNA | **9.42** | **1.71 ×** | 3.844 | 73.440 | 0.627 | 0.806 | 4.257 |

*Table 5.* Full attention vs. SNA at ×4 SR (RealPhoto60).

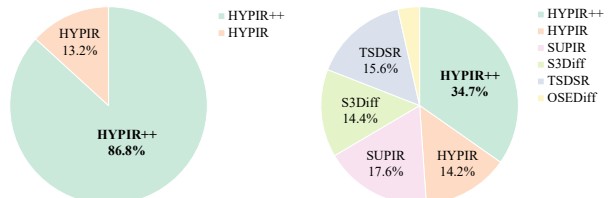

*Figure 10.* Results of two user studies. Left: HYPIR vs. HYPIR++. Right: HYPIR++ vs. other state-of-the-art methods.

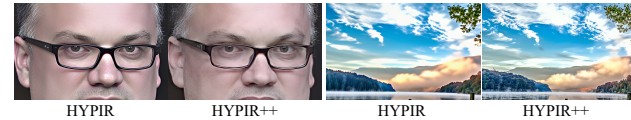

*Figure 11.* Visual comparison between HYPIR and HYPIR++ under the cascade setting; HYPIR exhibits noticeable over-saturation.

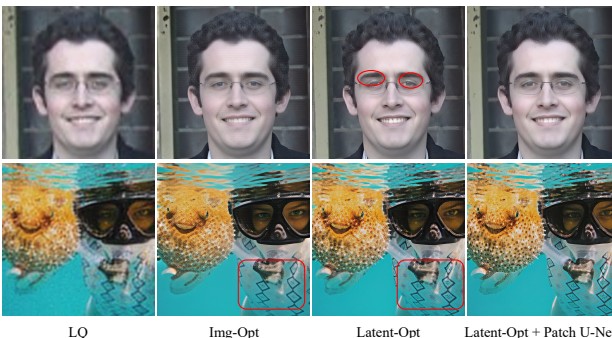

| LQ | Img-Opt | Latent-Opt | Latent-Opt + Patch U-Net |

*Figure 12.* Visual comparison on optimization space. Transitioning from image space to latent space yields more details. Adding the latent patch U-Net further enhances local structures and reduces artifacts. All models run for a single inference step.

## 6. Conclusion

We introduce HYPIR++, a framework dedicated to two-stage cascaded large-factor real-world super-resolution (i.e., total upscaling factor $\geq 4\times$, with $8\times$ as a natural extension). HYPIR++ addresses the three practical issues of a naive HYPIR cascade: removing the degradation removal encoder and noise augmentation better preserves fidelity cues across stages; latent-space adversarial training with our latent ConvNeXt and latent patch discriminator restores finer details and more stable local structures; and replacing full attention with sparse neighbor attention enables efficient high-resolution inference without tiling. Under the same two-stage cascade, HYPIR++ delivers higher perceptual quality with a $1.71\times$ speedup over HYPIR, offering a more practical solution for large-factor real-world SR.

**Limitations.** HYPIR++ is a cascade-aware solution targeted at large-factor SR ($\geq 4\times$) rather than a universal replacement for HYPIR: in small-factor settings (e.g., $2\times$ or $3\times$) a single HYPIR step already suffices, so a two-stage cascade is not a representative use case. Furthermore, our cascade reuses the same generator at both stages without stage-specific specialization, and a learned stage-aware variant is left to future work.

## Acknowledgement

This work was supported by National Natural Science Foundation of China (Grant No.62276251); in part by Macau Science and Technology Development Fund under 001/2024/SKL, 0119/2024/RIB2 and 0110/2025/R1B2; in part by Research Committee at University of Macau under MYRG-CRG2025-00031-FST and MYRG-GRG2025-00086-FST; in part by the Guangdong Basic and Applied Basic Research Foundation under Grant 2024A1515012536.

## Impact Statement

This paper advances the field of machine learning for real-world image super-resolution, with potential applications spanning consumer photo restoration, medical and remote-sensing imagery, and accessibility tools that present low-quality visual content at higher resolution. By delivering large-factor restoration at $1.71\times$ the speed of the cascaded baseline, HYPIR++ also reduces the inference-time computational and energy cost of practical deployment.

We note, however, that HYPIR++—like other diffusion-prior-based restoration methods—synthesizes high-frequency details from a learned generative prior rather than recovering information that is genuinely present in the low-quality input. The predicted details are therefore plausible but not faithful, and outputs should not be treated as ground truth for downstream tasks where detail-level accuracy carries real-world consequences. In particular, restored images are unsuitable as forensic or legal evidence (e.g., for identifying individuals or objects in degraded surveillance footage), and care should be taken before applying the model to medical, scientific, or safety-critical imagery without additional domain-specific validation.

Finally, because HYPIR++ builds on a pretrained text-to-image diffusion backbone (FLUX), it may inherit biases present in the underlying training data—including imbalanced representation across demographic groups, scene types, and visual styles. Practitioners deploying the model in user-facing settings should evaluate fairness on their target distribution and adopt safeguards against the misuse of generative restoration to fabricate visual content.

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

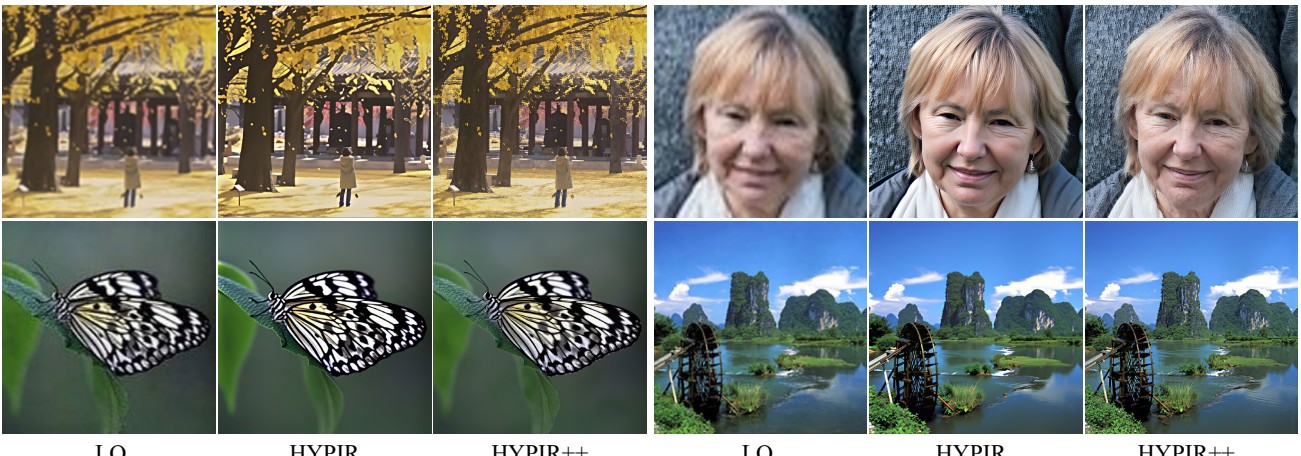

| LQ | HYPIR | HYPIR++ | LQ | HYPIR | HYPIR++ |

*Figure 13.* Visual comparison between HYPIR and HYPIR++ on RealPhoto60. Compared against HYPIR, HYPIR++ generates images with more fine-grained details, while also achieving more natural and well-balanced color saturation.

## A. Comparison Between HYPIR and HYPIR++

As shown in Fig. 13, HYPIR on RealPhoto60 exhibits noticeable over-saturation (e.g., butterfly, leaves) and misses fine details (e.g., hair, grass), whereas HYPIR++ recovers richer details with more natural color balance.

## B. More Qualitative Comparison

Visual comparisons of different methods on RealPhoto60 (Yu et al., 2024), RealSR (Cai et al., 2019) and DRealSR (Wei et al., 2020) are shown in Fig. 14, Fig. 15, and Fig. 16. Specifically, for Fig. 14, HYPIR++ effectively restores details of the eaves, trees, and the woman wearing a trench coat. For Fig. 15, HYPIR++ recovers the structural textures on the backpack. As illustrated in Fig. 16, compared with other methods, HYPIR++ produces noticeably richer and more visually appealing details (e.g., flowers, leaves). Overall, the visual comparisons demonstrate that HYPIR++ removes image degradations and recovers structural and sharp details across diverse real-world scenes.

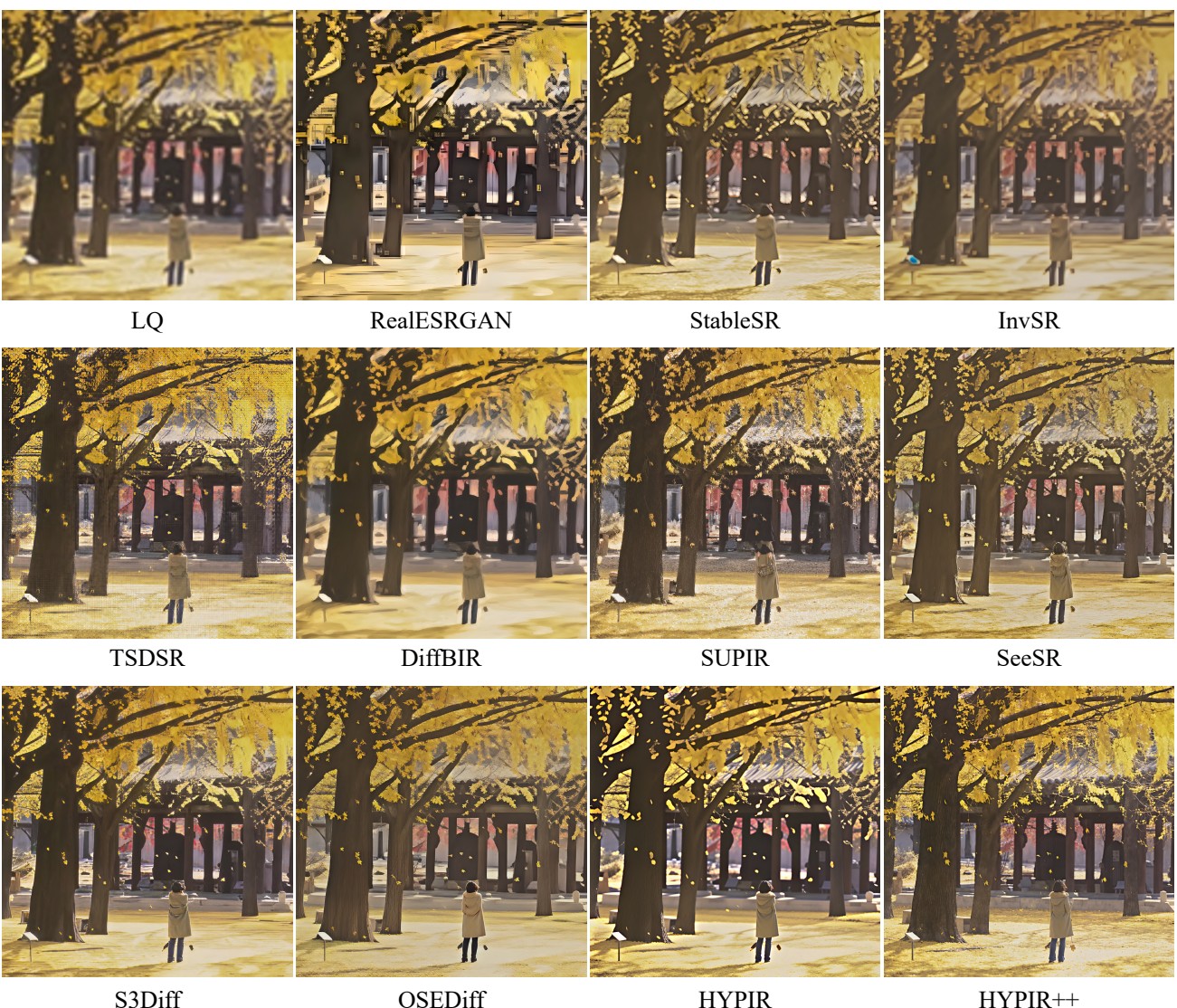

*Figure 14.* Visual comparison on RealPhoto60. Compared with other methods, HYPIR++ can effectively restore details of the eaves, trees, and the woman wearing a trench coat.

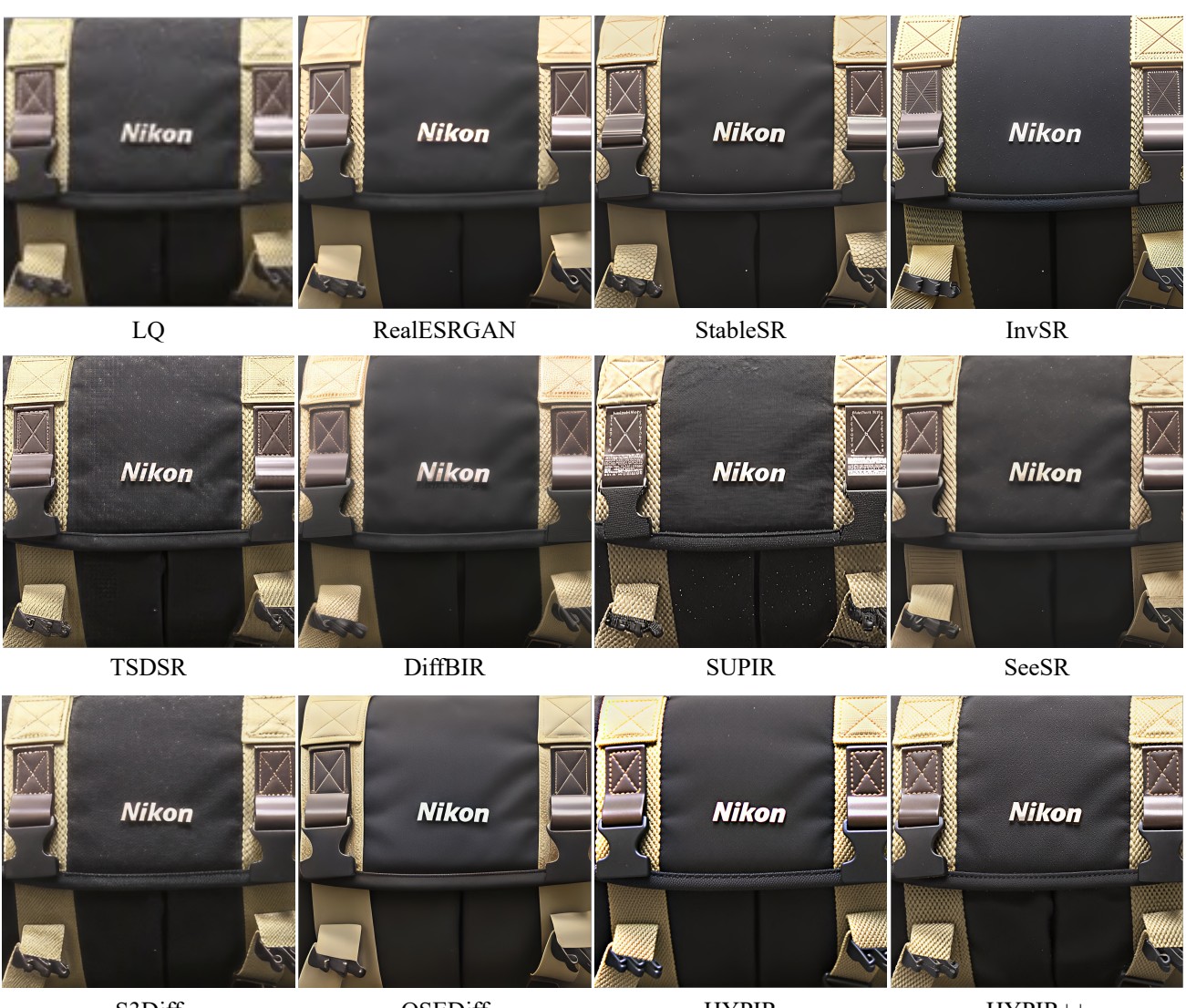

LQ  RealESRGAN  StableSR  InvSR

TSDSR  DiffBIR  SUPIR  SeeSR

S3Diff  OSEDiff  HYPIR  HYPIR++

*Figure 15.* Visual comparison on RealSR: HYPIR++ better restores the structural textures of the backpack.

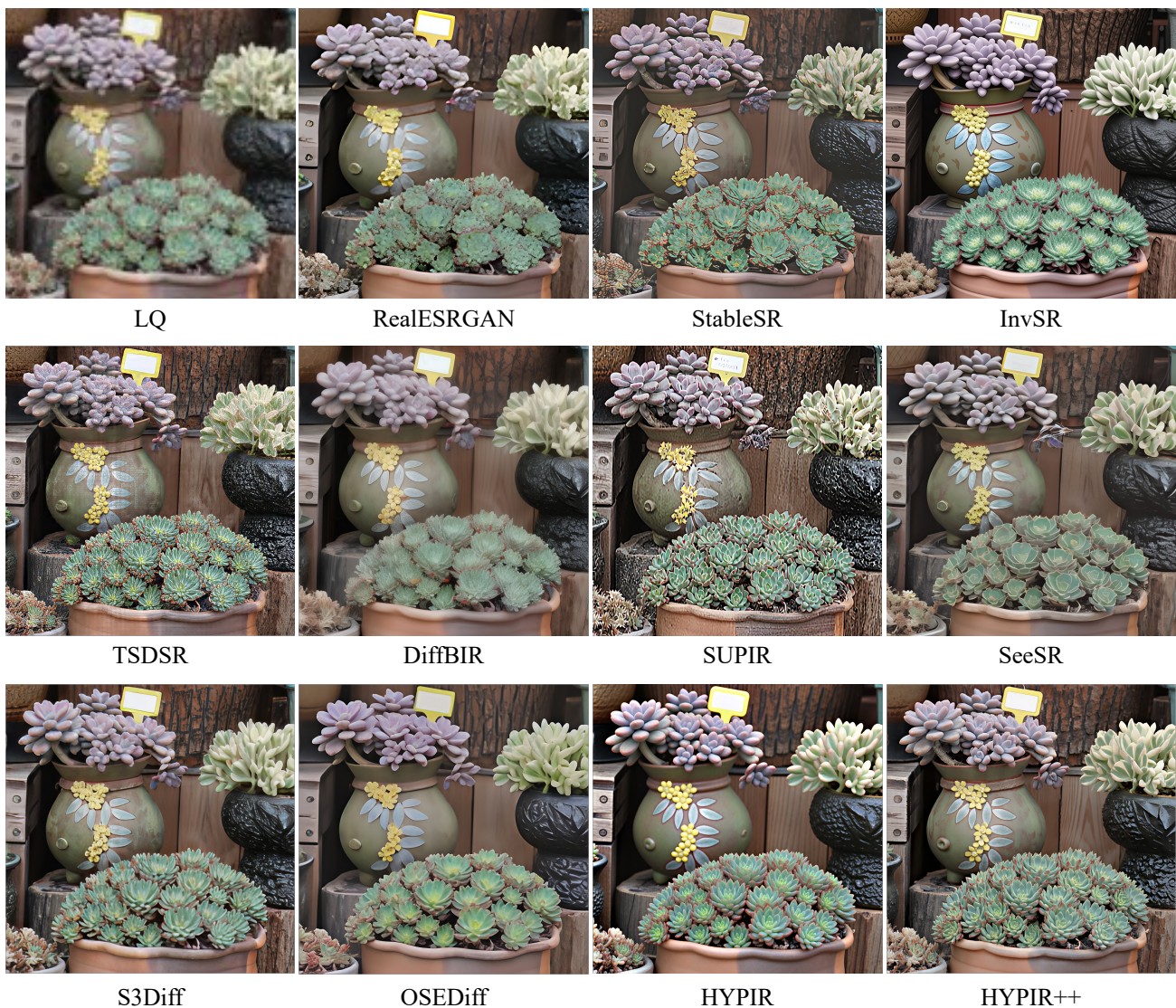

*Figure 16.* Visual comparison on DRealSR: HYPIR++ produces noticeably richer and more visually appealing details.

