# OpenReview forum: "Enhanced Latent-Space Adversarial Training for Super-Resolution"
_ICML.cc/2026/Conference — ICML 2026 regular_

### Official Review · Reviewer_zhDV · 2026-03-07

**Soundness:** 2
**Presentation:** 3
**Significance:** 3
**Originality:** 2
**Overall Recommendation:** 4
**Confidence:** 4

**Summary:**

The paper presents HYPIR++, an enhanced framework for real-world image super-resolution based on single-step diffusion models. Building upon HYPIR, it mitigates color oversaturation by removing the degradation removal encoder. It shifts adversarial training to the latent space using a distilled latent ConvNeXt and a latent Patch U-Net discriminator. Additionally, it reduces inference latency to enable tile-free SR by shortening text sequences and introducing a Sparse Neighbor Attention (SNA) mechanism.

**Compliance With Llm Reviewing Policy:**

Affirmed.

**Final Justification:**

The rebuttal addresses most of my concerns. Though the novelty and contribution are relatively limited, the experimental results are good from an engineering and practical perspective. Therefore, I raise my score from 3 to 4.

**Key Questions For Authors:**

1.	What are the specific algorithmic or structural differences between the proposed SNA and the cited Block Sparse Attention?
2.	Could the submission provide a theoretical explanation to justify why the architecture in Figure 3(c) effectively guides the generator better than Figure 3(b)?
3.	Section 5.1 mentions evaluating with PSNR, SSIM, and LPIPS, why these metrics are missing from the results?

**Limitations:**

The submission does not sufficiently discuss limitations. It is recommended to add a dedicated paragraph addressing technical failure cases.

**Strengths And Weaknesses:**

Strengths:
The method effectively addresses the computational bottleneck caused by full attention in diffusion-based SR. The use of knowledge distillation to construct a specialized latent ConvNeXt discriminator is a creative highlight. This cleverly enables effective latent feature extraction under strict VRAM constraints. The visual comparisons effectively demonstrate the framework's clear improvements in artifact removal and natural texture recovery.

Weaknesses:
1. Limited fundamental algorithmic innovation: the proposed SNA appears to be a simple modification of the cited Block Sparse Attention (Guo et al., 2024). The PatchGAN-based discriminator is a common practice. Meanwhile, reducing the text sequence length leans more toward a standard engineering optimization technique.
2. Section 5.1 states that synthetic datasets are evaluated using full-reference metrics (PSNR, SSIM, LPIPS), yet these metrics are completely absent from all results tables, leaving the Fidelity to LQ Input without quantitative support.
3. In Section 4.2, the paper compares two architectures (Figure 3b and 3c) and selects the latter based on experiment observations. Given the absence of theoretical justification here, this empirical comparison should be placed in the “Ablation Studies” section rather than the core methodology section.

---

> ### Author Rebuttal · Authors · 2026-03-31
>
> We sincerely appreciate your insightful feedback. Below are our detailed responses.
>
> `[W1]: Limited algorithmic innovation`
>
> We acknowledge that SNA builds upon Block Sparse Attention. The key adaptations include: (1) treating text as a single block that always participates in attention, applying sparsity only to image-image interactions; (2) a two-stage mechanism — neighbor-restriction followed by top-k selection based on block-level feature similarity. These are specifically adapted for the text-image cross-modal setting in DiT-based restoration, enabling tile-free processing and achieving 1.71× speedup with negligible quality loss (Table 5).
>
> Similarly, while the patch discriminator and text token reduction build on existing techniques, each is motivated by a specific observation in our setting: the global latent ConvNeXt tends to overlook fine-grained local structures, necessitating the patch-based complement; and image-to-image attention dominates (~80%) in the FLUX MMDiT blocks (Fig. 5), indicating that detailed textual information has negligible influence on restoration.
>
> `[W2 & Q3]: Missing full-reference metrics.`
>
> We apologize for the confusion. Tables 1 correspond to ×8 SR, but RealSR and DRealSR only provide GT pairs up to ×4, so FR metrics cannot be computed for ×8 outputs.
>
> For the ×4 setting, the degradations in RealSR and DRealSR are relatively mild, so we directly evaluate all methods in a single-pass manner. Results are shown below:
>
> **RealSR ×4** (best in **bold**, second best with †):
> |Model|PSNR↑|SSIM↑|LPIPS↓|FID↓|NIQE↓|MUSIQ↑|MANIQA↑|CLIP-IQA↑|DEQA↑|
> |---|---|---|---|---|---|---|---|---|---|
> |Ours|22.72|0.636|**0.203**|**62.46**|**4.582**|72.77†|0.600†|0.677|**4.049**|
> |HYPIR|19.62|0.568|0.249|84.73|5.146|71.55|0.586|0.668|3.811|
> |DiffBIR|23.56†|0.599|0.265|82.01|5.370|72.36|**0.607**|0.686|3.980†|
> |RealESRGAN|**24.24**|**0.679**|0.256|103.12|5.077|66.73|0.415|0.479|3.390|
> |S3Diff|23.42|0.643†|0.213†|65.74†|4.781|71.64|0.511|0.665|3.817|
> |SUPIR|23.42|0.607|0.275|77.38|5.957|67.87|0.560|0.689†|3.928|
> |TSDSR|21.97|0.610|0.221|76.97|4.597†|**73.05**|0.558|**0.711**|3.932|
>
> **DRealSR ×4** (best in **bold**, second best with †):
>
> |Model|PSNR↑|SSIM↑|LPIPS↓|FID↓|NIQE↓|MUSIQ↑|MANIQA↑|CLIP-IQA↑|DEQA↑|
> |---|---|---|---|---|---|---|---|---|---|
> |Ours|23.22|0.622|**0.183**|**49.84**|**4.465**|72.41†|**0.635**|0.689†|**4.139**|
> |HYPIR|22.81|0.653†|0.189†|75.95|4.610|72.10|0.610|0.680|3.969|
> |DiffBIR|24.21†|0.587|0.245|69.17|5.271|72.35|0.615†|0.687|4.033†|
> |RealESRGAN|**24.69**|**0.671**|0.229|83.14|4.611|67.46|0.424|0.502|3.468|
> |S3Diff|23.75|0.620|0.193|56.65†|4.593|72.06|0.533|0.671|3.954|
> |SUPIR|24.20|0.603|0.241|62.14|5.583|68.75|0.607|0.682|4.035|
> |TSDSR|22.51|0.592|0.193|56.51|4.572†|**72.97**|0.557|**0.704**|4.013|
>
>
> HYPIR++ achieves the best LPIPS and FID on both datasets, indicating superior perceptual quality with good fidelity. Methods with the highest PSNR/SSIM (e.g., RealESRGAN) produce oversmoothed results, reflecting the perception-distortion trade-off. Furthermore, IQA metrics cannot fully capture human perception. Our user studies (Sec. 5.3, Fig. 9) show 86.8% preference for HYPIR++ over HYPIR and 34.7% overall preference among all methods
>
> `[W3]: Architecture comparison placement.`
>
> Thanks for your suggestion. We will move the comparison to the ablation studies section in the revised manuscript.
>
> `[Q1]: SNA vs block sparse attention.`
>
> Please refer to our response to W1 above.
>
> `[Q2]: Justification for Fig. 3(c) over Fig. 3(b).`
>
> Both architectures converge well during knowledge distillation (Fig. 4), yet only the backbone-modification variant (Fig. 3c) effectively guides the generator during GAN training. We attribute this to the redundant upsampling-downsampling process in Fig. 3(b): the latent is first upsampled from 64×64 to 512×512, then progressively downsampled by the ConvNeXt backbone. This expansion and compression process loses high-frequency details in the latent representations, weakening the discriminator's ability to distinguish subtle quality differences between real and generated latents. In contrast, Fig. 3(c) directly processes the 64×64 latent by removing the downsampling operations in Stem and Stage0, better preserving the discriminative feature quality critical for adversarial training.
>
> `Limitations`
>
> We will add a dedicated limitations paragraph in the revised manuscript:
>
> - **Hallucination risk**: HYPIR++ may produce content inconsistent with the original image under extreme degradations (e.g., text, digits), a shared challenge among diffusion-based SR methods.
> - **Severe degradations**: The two-step cascade may still be insufficient under extremely severe degradations.
> - **Cascade overhead**: The cascade doubles inference computation. Although we achieve a 1.71× speedup, the overall latency may still limit applicability in latency-sensitive scenarios.

---

> > ### Author Rebuttal · Reviewer_zhDV · 2026-04-03
> >
> > The rebuttal is helpful and clarifies several implementation/experimental details. I appreciate the added evidence. However, the main concerns that affected my original overall assessment — especially regarding the strength/novelty of the contribution — remain only partially resolved. Therefore I keep my overall score unchanged.

---

> > > ### Author Response · Authors · 2026-04-03
> > >
> > > We would like to reiterate that our contribution is problem-driven: we address the challenges of cascading single-step diffusion models for real-world SR. While individual components build on existing techniques, diagnosing the root causes and developing effective solutions required careful investigation — for example, we trace over-saturation to the degradation-removal encoder amplifying the Flux generative bias (validated by ablation), and seemingly natural latent discriminator designs (Fig. 3b) fail while only carefully structured variants (Fig. 3c) succeed. We also validate these findings across backbones (FLUX and SD3). We respectfully believe the combination of strong empirical results (best LPIPS and FID on both datasets), practical impact (1.71× speedup), and these design insights constitutes a meaningful contribution.

---

### Official Review · Reviewer_BhWM · 2026-03-10

**Soundness:** 3
**Presentation:** 3
**Significance:** 3
**Originality:** 2
**Overall Recommendation:** 4
**Confidence:** 3

**Summary:**

This paper introduce HYPIR++, which enhances HYPIR with a latent adversarial training.
In detail, they propose a latent ConvNeXt by distlling the original pixel-level ConvNeXt into a latent one to use it as discriminator.
With the latent ConvNeXt discriminator, they additionally use patch U-Net discriminator to enhance the fine-details in the generated images.
Furthermore, to reduce the inference cost, they lower the number of attention operation by decreasing the text lengths from the obeservation that visual tokens T2I diffusion tends to attend itself and introduce sparse neighborhood attention.
Experiments show that the proposed method enhances various IQA metrics and quality of generated images, proving it by user study.

**Compliance With Llm Reviewing Policy:**

Affirmed.

**Final Justification:**

The authors' response addresses most of my concerns. However, I still have some remaining concern regarding novelty. So, I maintain my original rating, weak accept.

**Key Questions For Authors:**

First of all, please refer to the weakness section.

- In the first paragraph in section 4.3, the authors reduce the sequence length of text tokens. How this token reduction is performed? It should be elaborated in detail, if it reduce the actual inference cost largely.
- Did authors try the different architecture for the latent discriminator, except Convnext? I wonder that the training the latent convnext is significant contribution of this paper or not. For example, as E-LatentLPIPS requires latent VGG network, did authors test it as the model for discriminator?
- Why does the optimization in the latent space report the better performance than pixel space? Since IR requires the precise correction of pixels, the artifacts that autoencoder introduces may incurs the negative effects on IR model.

**Limitations:**

They do not discuss the limitations.

**Strengths And Weaknesses:**

## Strengths
- In a quantitative results, it surpasses the prior baseline HYPIR in both the image quality assessment metric, user study, and efficiency.
- Experiments about the latent convnext is interesting, simple l2 distillation while ablating the input stems achieves significantly better generation quality.
- Qualitative comparison shows that the proposed method synthesize the precise fine textures with the less distortion of original content and the balanced color saturation.

## Weaknesses
- All of the introduced components are somewhat incremental. For example, adversarial learning in the latent space of VAE is already explored in the one-step distllation papers such as LADD [A] and Diffusion2GAN [B]. Similarly, the sparse attention is also introduced by prior work (block sparse attention). Although the authors achieves an improvement in IR task, individual elements are not the contribution of this paper. This makes the contribution somewhat lower.

[A] Sauer, Axel, et al. "Fast high-resolution image synthesis with latent adversarial diffusion distillation." SIGGRAPH Asia 2024 Conference Papers. 2024.
[B] Kang, Minguk, et al. "Distilling diffusion models into conditional gans." European Conference on Computer Vision. Cham: Springer Nature Switzerland, 2024.



- Lack of motivation and ablation study.
There is no justification about why does the latent-level adversarial loss is better than pixel-level one. Intuitively, as the task of IR needs to adjust the fine details, it can be natural to modify it in pixel-space , where there is no errors in fine details that VAE introduces.
Also, there is no ablation study about the degradation removal encoder, which is one of the major difference compared to HYPIR.


- Prior method except HYPIR utilize the backbone diffusoin model different from FLUX, which is the model this paper used. It can be seen as minor issue, but the generation performance of backbone affects the performance largely, so it would be great to show the analysis on the effects of backbone model.

---

> ### Author Rebuttal · Authors · 2026-03-31
>
> We sincerely appreciate your insightful feedback. Below are our detailed responses.
>
> `[W1]: Incremental components (LADD/Diffusion2GAN Precedent)`
>
> We acknowledge that individual components build on existing techniques. However, under complex real-world degradations, single-step diffusion models are insufficient and cascade becomes necessary, introducing coupled issues (over-saturation, limited details, high latency). Our work provides a systematic, problem-driven solution.
>
> Each component targets a specific identified problem rather than combining techniques for incremental gain: removing the degradation-removal encoder resolves over-saturation; the latent ConvNeXt and patch U-Net discriminators enable adversarial training in latent space; and SNA enables tile-free high-resolution inference.
>
> `[W2.1 & Q3]: Why latent-space optimization outperforms pixel-space.`
>
> We empirically compare latent-space and pixel-space optimization in Table 3 and Fig. 11. Transitioning to latent space consistently improves all metrics (Table 3), and eliminates the grid-like artifacts produced by pixel-space optimization (Fig. 11). We attribute this to the shortcut introduced by the VAE decoder in pixel-space GAN training: since fake samples pass through the decoder while real samples (natural GT images) do not, the discriminator can exploit decoder-specific patterns (e.g., grid artifacts in Fig. 11) rather than learning meaningful quality differences. In latent-space GAN training, this shortcut is eliminated as both real and fake samples reside in the latent space without decoder interference.
>
> Regarding the concern that VAE errors may negatively affect restoration: the FLUX VAE has very high reconstruction accuracy. Empirically, latent-space optimization improves not only perceptual quality but also fidelity (Table 3), indicating that VAE errors are not a limiting factor.
>
>
> `[W2.2]: Missing ablation for degradation-removal encoder`
>
> We provide the quantitative ablation for the degradation-removal encoder on RealSR and DRealSR under the ×4 single-pass setting. It can be observed that removing the degradation-removal encoder improves LPIPS, FID, and the majority of NR-IQA metrics on both datasets.
>
> **RealSR ×4:**
> |Model|LPIPS↓|FID↓|NIQE↓|MUSIQ↑|MANIQA↑|CLIP-IQA↑|DEQA↑|
> |---|---|---|---|---|---|---|---|
> |HYPIR++|**0.203**|**62.46**|**4.582**|**72.77**|0.600|**0.677**|**4.049**|
> |HYPIR++ (w/ deg. enc.)|0.221|72.73|5.030|71.54|**0.577**|0.632|3.946|
>
> **DRealSR ×4:**
> |Model|LPIPS↓|FID↓|NIQE↓|MUSIQ↑|MANIQA↑|CLIP-IQA↑|DEQA↑|
> |---|---|---|---|---|---|---|---|
> |HYPIR++|**0.183**|**49.84**|**4.465**|**72.41**|0.635|**0.689**|**4.139**|
> |HYPIR++ (w/ deg. enc.)|0.209|62.35|4.732|70.54|**0.644**|0.642|3.950|
>
> `[W3]: Effect of backbone model`
>
> Since HYPIR built on FLUX has demonstrated strong restoration capability, we naturally build upon it for further exploration. The HYPIR vs HYPIR++ comparison is entirely fair as both use the identical FLUX backbone. To validate that our improvements are not backbone-specific, we provide results on an SD3-based model:
>
> |Dataset|Model(SD3)|NIQE↓|MUSIQ↑|MANIQA↑|CLIP-IQA↑|DEQA↑|
> |---|---|---|---|---|---|---|
> |RealPhoto60|HYPIR|3.858|69.07|0.447|0.760|4.195|
> |RealPhoto60|HYPIR++|**3.316**|**71.07**|**0.457**|**0.775**|**4.231**|
> |DRealSR×4|HYPIR|5.315|**72.22**|**0.590**|0.691|3.761|
> |DRealSR×4|HYPIR++|**4.762**|72.14|0.585|**0.708**|**3.852**|
> |RealSR×4|HYPIR|5.838|**72.61**|0.593|0.713|3.800|
> |RealSR×4|HYPIR++|**4.980**|72.45|**0.595**|**0.722**|**3.924**|
>
> HYPIR++ (SD3) outperforms HYPIR (SD3) on the majority of metrics, suggesting our design principles can generalize to other diffusion-based restoration frameworks..
>
> `[Q1]: Text token reduction details`
>
> Our analysis (Fig. 5) shows image-to-image interaction dominates (~80%) attention in FLUX MMDiT blocks, indicating textual granularity has negligible influence. We simply limit the VLM caption length, reducing text tokens from 512 to 128.
>
> `[Q2]: Alternative latent discriminator architectures`
>
> We did not explore other architectures. ConvNeXt already demonstrated strong performance in HYPIR, so we focused on adapting it to latent space via knowledge distillation. The backbone-modification architecture (Fig. 3c) proved critical. We consider exploring alternatives as future work.
>
> `Limitations`
>
> We will add a dedicated limitations paragraph in the revised manuscript:
> - **Hallucination risk**: HYPIR++ may produce content inconsistent with the original image under extreme degradations (e.g., text, digits), a shared challenge among diffusion-based SR methods.
> - **Severe degradations**: The two-step cascade may still be insufficient under extremely severe degradations.
> - **Cascade overhead**: The cascade doubles inference computation. Although we achieve a 1.71× speedup, the overall latency may still limit applicability in latency-sensitive scenarios.

---

> > ### Author Rebuttal · Reviewer_BhWM · 2026-04-03
> >
> > The authors' response addresses most of my concerns. However, I still have some remaining concern regarding novelty. While I understand the authors' argument that each component is introduced to resolve a specific problem, the overall contribution still appears somewhat incremental to me. Therefore, I will maintain my score at weak accept.

---

> > > ### Author Response · Authors · 2026-04-03
> > >
> > > We sincerely thank the reviewer for acknowledging our response and for the constructive feedback throughout the review process. We will incorporate the suggested improvements in the revised manuscript.

---

### Official Review · Reviewer_3rD9 · 2026-03-13

**Soundness:** 3
**Presentation:** 3
**Significance:** 3
**Originality:** 2
**Overall Recommendation:** 3
**Confidence:** 4

**Summary:**

This paper addresses real-world image super-resolution and proposes HYPIR++, an improved framework built upon the HYPIR model. The authors aim to mitigate several limitations observed in HYPIR, including over-saturation artifacts, insufficient fine-grained details, and relatively high inference latency. To address these issues, the paper introduces several design modifications, including latent-space adversarial training with a ConvNeXt-based discriminator, a latent patch discriminator to improve local structures, and a sparse neighbor attention mechanism to reduce computational complexity. Extensive experiments on real-world super-resolution benchmarks demonstrate improved perceptual quality and user preference compared to existing approaches, along with faster inference.

**Compliance With Llm Reviewing Policy:**

Affirmed.

**Final Justification:**

My main concern regarding originality remains only partially addressed. While the rebuttal highlights non-trivial design choices and diagnostic insights, the proposed solutions still largely rely on existing techniques applied in a predictable manner. As such, the level of conceptual novelty and broader impact beyond this specific system remains somewhat limited.

The rebuttal partially addresses my concerns, but it does not significantly change my overall assessment, and therefore my score remains unchanged.

**Key Questions For Authors:**

1. **Novelty clarification.**
Many of the components used in the proposed system appear to rely on existing ideas (e.g., adversarial training, patch discriminators, sparse attention). Could the authors clarify what they consider to be the primary methodological novelty beyond the integration of these components within the HYPIR framework?

2. **Generalization to other diffusion-based restoration models.**
The proposed improvements are implemented within the HYPIR architecture. Do the authors expect the same design principles (e.g., latent-space adversarial training and patch discriminators) to generalize to other diffusion-based image restoration models?

3. **Analysis of design choices.**
It would be helpful to better understand why latent-space adversarial training improves perceptual quality in this setting. Do the authors have further insights or analysis that could help explain this behavior?

**Limitations:**

No. The paper does not explicitly discuss the limitations of the proposed method or potential societal impacts. It would strengthen the paper if the authors could discuss scenarios where the method may fail (e.g., extremely severe degradations).

**Strengths And Weaknesses:**

**Strengths**

- Clear motivation and problem formulation. The paper clearly identifies limitations of the previous HYPIR framework and motivates the proposed improvements in a practical and understandable way.

- Well-written and easy to follow. The manuscript is clearly structured and the technical descriptions are generally easy to understand. The overall narrative and presentation are strong.

- Solid experimental evaluation. The authors provide comprehensive empirical results, including comparisons with several baselines, perceptual metrics, and ablation studies. The improvements appear consistent across multiple evaluation metrics and user studies.

**Weaknesses**

- Limited conceptual novelty. Many of the individual components used in the proposed framework rely on existing techniques, such as adversarial training, patch-based discriminators, and sparse attention mechanisms. The contribution appears to lie primarily in combining and refining these ideas within the HYPIR framework, rather than introducing a fundamentally new methodological concept.

- Incremental extension of prior work. Since the proposed approach builds directly on the existing HYPIR framework, several design changes appear to be incremental improvements rather than substantial methodological innovations.

- Limited discussion of broader insights. The paper mainly focuses on improving system performance, but provides limited analysis on why the proposed modifications are particularly effective, or whether the design principles could generalize to other diffusion-based restoration frameworks.

---

> ### Author Rebuttal · Authors · 2026-03-31
>
> We sincerely appreciate your insightful feedback. Below are our detailed responses.
>
> `[W1 & W2 & Q1]: Limited conceptual novelty / incremental extension`
>
> We acknowledge that the individual components build on existing techniques. However, we believe the value of this work lies in addressing a practical challenge in real-world SR: under complex degradations, single-step diffusion models struggle to simultaneously remove degradations and generate high-quality details, and cascading the model improves quality but introduces coupled issues (over-saturation, limited fine details, high latency). Our work provides a systematic, problem-driven solution validated by comprehensive experiments.
>
> Each component targets a specific identified problem rather than combining techniques for incremental gain: removing the degradation-removal encoder resolves over-saturation in cascading (quantitatively validated in our new ablation); the latent ConvNeXt discriminator is designed to enable effective adversarial training in latent space under VRAM constraints — with only the backbone-modification architecture (Fig. 3c) proving effective while the intuitive variant (Fig. 3b) fails; and SNA is specifically adapted for the text-image cross-modal setting in DiT-based restoration, enabling tile-free high-resolution inference.
>
>
> `[W3.1 & Q3]: Why latent-space adversarial training improves perceptual quality in this setting?`
>
> We empirically compare latent-space and pixel-space optimization in Table 3 and Fig. 11. Transitioning to latent space consistently improves all metrics (Table 3), and eliminates the grid-like artifacts produced by pixel-space optimization (Fig. 11). We attribute this to the shortcut introduced by the VAE decoder in pixel-space GAN training: since fake samples pass through the decoder while real samples (natural GT images) do not, the discriminator can exploit decoder-specific patterns (e.g., grid artifacts in Fig. 11) rather than learning meaningful quality differences. In latent-space GAN training, this shortcut is eliminated as both real and fake samples reside in the latent space without decoder interference.
>
> `[W3.2 & Q2]: Generalization to other diffusion-based restoration models.`
>
> To validate the generalizability of our design principles, we apply latent-space optimization to an SD3-based version of HYPIR. Results on three datasets are shown below:
>
> |Dataset|Model(SD3)|NIQE↓|MUSIQ↑|MANIQA↑|CLIP-IQA↑|DEQA↑|
> |---|---|---|---|---|---|---|
> |RealPhoto60|HYPIR|3.858|69.07|0.447|0.760|4.195|
> |RealPhoto60|HYPIR++|**3.316**|**71.07**|**0.457**|**0.775**|**4.231**|
> |DRealSR×4|HYPIR|5.315|**72.22**|**0.590**|0.691|3.761|
> |DRealSR×4|HYPIR++|**4.762**|72.14|0.585|**0.708**|**3.852**|
> |RealSR×4|HYPIR|5.838|**72.61**|0.593|0.713|3.800|
> |RealSR×4|HYPIR++|**4.980**|72.45|**0.595**|**0.722**|**3.924**|
>
> HYPIR++ (SD3) outperforms HYPIR (SD3) on the majority of metrics across all three datasets. These results suggest that our design principles can generalize to other diffusion-based restoration frameworks.
>
>
> `Limitations`
>
> We will add a dedicated limitations paragraph in the revised manuscript:
>
> - **Hallucination risk**: HYPIR++ may produce content inconsistent with the original image under extreme degradations (e.g., text, digits), a shared challenge among diffusion-based SR methods.
> - **Severe degradations**: The two-step cascade may still be insufficient under extremely severe degradations.
> - **Cascade overhead**: The cascade doubles inference computation. Although we achieve a 1.71× speedup, the overall latency may still limit applicability in latency-sensitive scenarios.

---

> > ### Author Rebuttal · Reviewer_3rD9 · 2026-04-02
> >
> > While the problem-driven perspective is clearer, the limitations of the naive cascading baseline appear largely expected, and the proposed solutions rely on well-established techniques applied in a fairly predictable manner. As such, questions remain regarding conceptual novelty, and it is unclear what new insights or broader understanding this work provides or how it advances the field beyond this specific system.

---

> > > ### Author Response · Authors · 2026-04-03
> > >
> > > We show that directly cascading single-step diffusion SR models introduces specific issues — over-saturation from the generative prior being reinforced across stages, and loss of fine details due to pixel-space optimization limitations. While the existence of these issues may appear expected, diagnosing their root causes and developing effective solutions required
> > >   careful investigation. For instance, we trace over-saturation to the degradation-removal encoder and noise augmentation amplifying the Flux generative bias, and we find that seemingly natural latent discriminator designs (Fig. 3b) fail while only carefully structured variants (Fig. 3c) succeed.
> > >
> > > Regarding why latent-space optimization outperforms pixel-space: we observe consistent improvements across all metrics (Table 3) and elimination of grid-like artifacts (Fig. 11). One plausible explanation is that in pixel-space GAN training, the discriminator can exploit decoder-specific patterns since fake samples pass through the VAE decoder while real samples do not. Latent-space training avoids this discrepancy. We validate this advantage on an SD3-based model (see W3), confirming it is not specific to the FLUX backbone.
> > >
> > > We will clarify these insights in the revision.

---

### Official Review · Reviewer_Nht3 · 2026-03-15

**Soundness:** 3
**Presentation:** 3
**Significance:** 2
**Originality:** 3
**Overall Recommendation:** 4
**Confidence:** 3

**Summary:**

This manuscript presents HYPIR++, an extended version of the recent diffusion-based single-step super-resolution (SR) model, HYPIR. To address the limitations of HYPIR when applied to real-world degradations (particularly in cascaded usage), the authors propose a series of improvements: (1) removing the degradation-removal encoder and noise augmentation; (2) employing a distilled Latent ConvNeXt and a patch-based UNet discriminator for fully latent-space adversarial optimization; (3) accelerating inference by shortening text prompts and applying a Sparse Neighbor Attention (SNA) mechanism. The method is evaluated on multiple real-world super-resolution benchmarks.

**Compliance With Llm Reviewing Policy:**

Affirmed.

**Final Justification:**

The rebuttal addresses most of my concerns.

**Key Questions For Authors:**

1. **Baseline Fairness:** Can you provide results for all models (including HYPIR++) evaluated in their standard, intended **single-pass** setting for Tables 1 and 2, rather than enforcing a cascade?
2. **Fidelity Metrics:** Please provide LPIPS and FID scores for the paired datasets (RealSR and DRealSR) relative to the ground-truth images.
3. **Missing Ablation:** Please provide a quantitative ablation table isolating the impact of removing the degradation-removal encoder and noise augmentation. How do the IQA metrics and LPIPS change with and without these modules?

**Limitations:**

The methodology of the paper involves the use of AI image generation technology, which could give rise to certain ethical concerns.

**Strengths And Weaknesses:**

# Strengths

1. **Practical Problem Setting:** Overall, this study addresses a relevant challenge regarding the high computational costs and memory bottlenecks associated with diffusion-based SR models. Eliminating the need for block-based tiling during high-resolution inference is a highly practical contribution.
2. **Visual Quality:** Qualitative results (Figures 7, 8, and 12) indicate that the proposed engineering improvements effectively reduce the oversaturation artifacts present in the original HYPIR model, yielding visually pleasing high-frequency details.
3. **Efficiency Improvements:** The integration of Sparse Neighbor Attention (SNA) and token truncation provides a considerable 1.71x inference speedup without severe perceptual quality degradation, a claim strongly supported by the ablation studies in Table 5.

# Weaknesses

### 1. Incremental Contribution Relative to Framing
The contribution appears more incremental than the paper's framing suggests. Relative to HYPIR, this work presents a collection of engineering modifications: removing two modules, moving losses to latent space, adding a latent discriminator, adding a patch discriminator, shortening text, and using sparse attention. While this can still be publishable, the paper overstates the conceptual leap. The scientific contribution is not a new learning principle; it is mainly a stronger recipe for a specific pipeline.


### 2. Inadequate Evaluation Metrics for Paired Datasets
The paper evaluates RealSR and DRealSR using *only* No-Reference Image Quality Assessment (NR-IQA) metrics (NIQE, MUSIQ, MANIQA, etc.). RealSR and DRealSR are **paired** datasets with ground-truth high-resolution images. Relying solely on NR-IQA is a major red flag, as generative models (especially GANs and diffusion models) can easily "game" these metrics by hallucinating realistic but unfaithful textures. The authors must report Full-Reference metrics—specifically FID (to measure distribution distance) and LPIPS (to measure perceptual fidelity relative to ground truth)—to substantiate the perception-distortion trade-off.

### 3. Missing Crucial Quantitative Ablations
The authors claim in Section 4.1 that removing the degradation-removal encoder and noise augmentation resolves the "oversaturation" issue. However, this headline claim is supported only by a single qualitative figure (Figure 10). There are no quantitative ablation studies isolating the impact of this removal on final metrics. In a rigorous machine learning conference, structural claims must be supported by systematic ablation studies.

### 4. Inconsistencies and Careless Errors
   **Typo/Inconsistency:** The caption of Table 1 contains a clear mathematical error. It states "$\times 4$ SR on the DRealSR, RealSR datasets (from $128 \times 128$ inputs)$", but scaling from 128 to 1024 is $\times 8$ upsampling. The text within the table writes "$\times 8$", contradicting the caption. Such careless inconsistencies undermine confidence in the paper's empirical rigor.

---

> ### Author Rebuttal · Authors · 2026-03-31
>
> We sincerely appreciate your insightful feedback. Below are our detailed responses.
>
> `[W1]:  Incremental contribution relative to framing.`
>
> We appreciate this feedback. We clarify that the core motivation of HYPIR++ is addressing real-world SR under complex degradations, where single-step diffusion models are insufficient and cascade becomes necessary. The cascade paradigm introduces three coupled issues (over-saturation, limited details, high latency), and our contribution is a systematic solution to make cascade-based real-world SR practical. We will revise the framing to more accurately reflect this positioning, avoiding overstating conceptual novelty.
>
> `[W2 & Q1 & Q2]: Full-reference metrics for paired datasets / single-pass comparison.`
>
> The results in our original Tables 1 correspond to ×8 SR on RealSR and DRealSR. **Since these datasets only provide ground-truth pairs up to ×4, the ×8 outputs have no corresponding GT, and full-reference metrics cannot be computed**. For the ×4 setting, the degradations in RealSR and DRealSR are relatively mild, so we directly evaluate all methods in a single-pass manner. Results are shown below:
>
> **RealSR ×4** (best in **bold**, second best with †):
>
>   | Model | LPIPS↓ | FID↓ | NIQE↓ | MUSIQ↑ | MANIQA↑ | CLIP-IQA↑ | DEQA↑ |
>   |-------|--------|------|-------|--------|---------|-----------|-------|
>   | Ours | **0.203** | **62.46** | **4.582** | 72.77† | 0.600† | 0.677 | **4.049** |
>   | HYPIR | 0.249 | 84.73 | 5.146 | 71.55 | 0.586 | 0.668 | 3.811 |
>   | DiffBIR | 0.265 | 82.01 | 5.370 | 72.36 | **0.607** | 0.686 | 3.980† |
>   | InvSR | 0.266 | 111.59 | 5.448 | 72.47 | 0.524 | 0.671 | 3.759 |
>   | OSEDiff | 0.258 | 94.06 | 4.790 | 71.11 | 0.477 | 0.635 | 3.715 |
>   | RealESRGAN | 0.256 | 103.12 | 5.077 | 66.73 | 0.415 | 0.479 | 3.390 |
>   | S3Diff | 0.213† | 65.74† | 4.781 | 71.64 | 0.511 | 0.665 | 3.817 |
>   | SUPIR | 0.275 | 77.38 | 5.957 | 67.87 | 0.560 | 0.689† | 3.928 |
>   | SeeSR | 0.245 | 84.46 | 5.078 | 71.85 | 0.568 | 0.659 | 3.904 |
>   | StableSR | 0.255 | 91.58 | 5.017 | 71.10 | 0.497 | 0.631 | 3.588 |
>   | TSDSR | 0.221 | 76.97 | 4.597† | **73.05** | 0.558 | **0.711** | 3.932 |
>
>   **DRealSR ×4** (best in **bold**, second best with †):
>
>   | Model | LPIPS↓ | FID↓ | NIQE↓ | MUSIQ↑ | MANIQA↑ | CLIP-IQA↑ | DEQA↑ |
>   |-------|--------|------|-------|--------|---------|-----------|-------|
>   | Ours | **0.183** | **49.84** | **4.465** | 72.41 | **0.635** | 0.689† | **4.139** |
>   | HYPIR | 0.189† | 75.95 | 4.610 | 72.10 | 0.610 | 0.680 | 3.969 |
>   | DiffBIR | 0.245 | 69.17 | 5.271 | 72.35 | 0.615† | 0.687 | 4.033† |
>   | InvSR | 0.243 | 89.42 | 5.135 | 72.50† | 0.529 | 0.674 | 3.810 |
>   | OSEDiff | 0.229 | 76.18 | 4.680 | 71.42 | 0.469 | 0.639 | 3.713 |
>   | RealESRGAN | 0.229 | 83.14 | 4.611 | 67.46 | 0.424 | 0.502 | 3.468 |
>   | S3Diff | 0.193 | 56.65† | 4.593 | 72.06 | 0.533 | 0.671 | 3.954 |
>   | SUPIR | 0.241 | 62.14 | 5.583 | 68.75 | 0.607 | 0.682 | 4.035 |
>   | SeeSR | 0.217 | 60.66 | 4.908 | 72.23 | 0.583 | 0.664 | 4.001 |
>   | StableSR | 0.229 | 71.44 | 4.897 | 71.31 | 0.501 | 0.661 | 3.597 |
>   | TSDSR | 0.193 | 56.51 | 4.572† | **72.97** | 0.557 | **0.704** | 4.013 |
>
>   As shown in the tables above, HYPIR++ achieves the best LPIPS and FID on both datasets, indicating that our method can generate outputs with superior perceptual quality while maintaining good fidelity to the ground truth.
>
> `[W3 & Q3]: Missing quantitative ablation for degradation-removal encoder.`
>
> We provide the requested quantitative ablation for the degradation-removal encoder on RealSR and DRealSR. From these two tables, it can be observed that removing the degradation-removal encoder improves LPIPS, FID, and the majority of NR-IQA metrics on both datasets.
>
> **RealSR ×4:**
> | Model | LPIPS↓ | FID↓ | NIQE↓ | MUSIQ↑ | MANIQA↑ | CLIP-IQA↑ | DEQA↑ |
> |-------|--------|------|-------|--------|---------|-----------|-------|
> | HYPIR++ | **0.203** | **62.46** | **4.582** | **72.77** | 0.600 | **0.677** | **4.049** |
> | HYPIR++ (with deg. encoder) | 0.221 | 72.73 | 5.030 | 71.54 | **0.577** | 0.632 | 3.946 |
>
> **DRealSR ×4:**
> | Model | LPIPS↓ | FID↓ | NIQE↓ | MUSIQ↑ | MANIQA↑ | CLIP-IQA↑ | DEQA↑ |
> |-------|--------|------|-------|--------|---------|-----------|-------|
> | HYPIR++ | **0.183** | **49.84** | **4.465** | **72.41** | 0.635 | **0.689** | **4.139** |
> | HYPIR++ (with deg. encoder) | 0.209 | 62.35 | 4.732 | 70.54 | **0.644** | 0.642 | 3.950 |
>
>
> `[W4]: Typo in table 1 caption`
>
>  Thanks for pointing the typo, we will correct it in the revised manuscript.
>
> `[Q1]: Single-pass evaluation`
>
> We provide ×4 single-pass results on RealSR and DRealSR in W2, where all methods are evaluated in their standard single-pass setting. HYPIR++ ranks among the top across the majority of metrics on both datasets.

---

> > ### Author Rebuttal · Reviewer_Nht3 · 2026-04-03
> >
> > The authors have largely addressed my concerns. Taking into account the comments from other reviewers and the authors' rebuttal, I have decided to maintain my original my score

---

### Decision · Program_Chairs · 2026-04-30

**Decision:**

Accept (regular)

**Comment:**

This paper receives final ratings of (4, 4, 4, 3). The reviewers agree that the paper addresses a well-motivated, practical challenge in real-world super-resolution by mitigating over-saturation and high latency in diffusion models. Despite concerns regarding the incremental nature of the technical contributions , the AC finds that the improvements including a 1.71x speedup, superior perceptual quality, and the insights into latent-space optimization justify acceptance.